# The Protective Effects of 2’-Fucosyllactose Against *E. Coli* O157 Infection Are Mediated by the Regulation of Gut Microbiota and the Inhibition of Pathogen Adhesion

**DOI:** 10.3390/nu12051284

**Published:** 2020-05-01

**Authors:** Yuanyifei Wang, Yan Zou, Jin Wang, Hui Ma, Bowei Zhang, Shuo Wang

**Affiliations:** 1Key Laboratory of Food Nutrition and Safety, Ministry of education, Tianjin University of Science and Technology, Tianjin 300457, China; wangyyf163@163.com; 2Tianjin Key Laboratory of Food Science and Health, School of Medicine, Nankai University, Tianjin 300071, China; 2120181361@mail.nankai.edu.cn (Y.Z.); wangjin@nankai.edu.cn (J.W.); mahui2018@mail.nankai.edu.cn (H.M.); bwzhang@nankai.edu.cn (B.Z.)

**Keywords:** 2’-fucosyllactose, *E. coli* O157, reduced colonization, gut microbiota, *Akkermansia*

## Abstract

As the richest component in human milk oligosaccharides (HMOs), 2’-fucosyllactose (2’-FL) can reduce the colonization of harmful microbiota in vivo, thus lowering the risk of infection; however, the mechanism for this is still unclear. In this study, a model of *Escherichia coli* O157 infection in healthy adult mice was established to explore the effect of 2’-FL intervention on *E. coli* O157 colonization and its protective effects on mice. The results showed that 2’-FL intake reduced *E. coli* O157 colonization in mice intestine by more than 90% (*p* < 0.001), and it also reduced intestinal inflammation, increased the content of fecal short-chain fatty acids, and enhanced intestinal barrier function. These beneficial effects were attributed to the increased expression of mucins such as *MUC2* (increased by more than 20%, *p* < 0.001), and inhibition of *E. coli* O157 cell adhesion (about 30% reduction, *p* < 0.001), and were associated with the modulation of gut microbiota composition. 2’-FL significantly increased the abundance of *Akkermansia*, a potential probiotic, which may represent the fundamental means by which 2’-FL enhances the expression of mucin and reduces the colonization of harmful bacteria. The current study may support the use of 2’-FL in the prevention of foodborne pathogen infections in human.

## 1. Introduction

2’-Fucosyllactose (2’-FL) is a kind of neutral, low-molecular-weight, human milk oligosaccharide (HMO), which is considered to play an extremely important role in host immunity [1]. The unique structure of 2’-FL allows it to resist digestion in the gastrointestinal tract, resulting in the majority readily reaching the large intestine for microbial decomposition, with a very small portion entering the blood and systemic circulation. 2’-FL has a molecular weight of 488.44 Da and is formed by the linking of fucose to lactose, with one end of the free semi-acetal hydroxyl group adopting a reductive structure. Compared with other milk oligosaccharides, 2’-FL is the most abundant type of HMO in human milk (with a proportion of 25%), as well as the only type that can be industrialized. The production of 2’-FL through successive chemical and microbial fermentation synthesis gained approval from the FDA in 2015 and 2016, respectively, and the European Union also ratified the inclusion of 2’-FL as a component of infant formula, a dietary supplement, and even as a nutraceutical [2]. Commercially, many companies have added 2’-FL to infant formula in an attempt to close the gap between formula and breast milk. Hence, research on the health benefits of 2’-FL as a novel dietary supplement has become a hotspot subject today.

The latest research demonstrates that the immunological function of suckling mice can be modulated by oral gavage with f 2’-FL as the only component, and that 2’-FL can also alleviate inflammation, enhance intestine barrier function, lower allergic reactions, and even improve the cognitive ability of mice [3,4,5,6,7]. Additionally, studies have also suggested that 2’-FL has protective effects against pathogen infection [8,9]. For example, based on in vitro studies, 2’-FL inhibited the adhesion of enteropathogenic *Escherichia coli* (EPEC), *Vibrio cholerae*, and even *Campylobacter jejuni* on intestinal epithelial cells (Caco-2), and reduced *Pseudomonas* spp. infection of the upper respiratory tract (A549) in humans [10,11,12,13]. In terms of the mechanism governing this, the most widely accepted theory is that 2’-FL acts as a soluble decoy receptor to block the invasion of pathogens [14,15,16]. However, it has also been reported that 2’-FL cannot reduce the adhesion of *Salmonella* spp. on Caco-2 cells, which suggests that the effect of 2’-FL in reducing pathogen adhesion may be selective, and not effective for all pathogens [17,18]. Research on the reduction in colonization by 2’-FL in vivo mainly focuses on clinical research in infants, for example, in a previous study demonstrating that feeding infants with formula milk containing 2’-FL significantly prevented diarrhea caused by *Campylobacter* spp. [19,20]. Similarly, some other short-chain carbohydrates can also reduce the colonization of pathogens in vivo [21]. For example, intake of fructo-oligosaccharides can reduce the colonization of *Salmonella* spp. in rats and galacto-oligosaccharide/polydextrose can alleviate diarrhea caused by *C. difficile* [22,23]. This was due to short-chain carbohydrates affecting the regulation of the intestinal microbiome. Multiplying beneficial bacteria compressed the living space and nutrition supply of harmful bacteria, thereby reducing colonization by pathogens. In recent years, many studies on 2’-FL regulation of intestinal microbiome have emerged. Studies have shown that 2’-FL can specifically stimulate proliferation of certain types of *Bifidobacteria* spp. and play a very important role in establishing a healthy intestinal microorganism environment for infants and children [24,25,26]. However, no studies have confirmed whether 2’-FL can reduce pathogen colonization in vivo by regulating intestinal microbiome.

As a kind of common pathogenic microorganism, enterohemorrhagic *Escherichia coli* (EHEC) is a subtype of *Escherichia coli*, typed by O antigen, and the main pathogenic strain is O157:H7. It can cause infectious diarrhea and hemorrhagic enteritis, to which infants and adults are both susceptible [27,28,29]. The present study aims to investigate the protective effect of 2’-FL against *E. coli* O157:H7 infection in mice model. In addition to the modulation of gut microbiota, its health benefits were also explored by examining its impact on enhancing the gut barrier and inhibiting harmful bacteria colonization.

## 2. Materials and Methods

### 2.1. Animals and Experimental Design

Thirty specific-pathogen-free (SPF) male mice (6 weeks old) were purchased from SPF Biotechnology Co., Ltd. (Beijing, China) and housed under constant temperature and humidity, with a 12 h/12 h light/dark cycle. Mice were fed a standard diet and were randomly divided into three groups. The experimental group (FL group) was intragastrically administrated with 0.05 g 2’-FL (Glycarbo Co. Ltd., Takamatsu, Japan)/100 g of body weight, which was calculated according to the total daily dose of HMOs an infant receives from breast milk. The model control group (MC group) and blank group (CK group) only received the same volume of PBS. The whole process of gavage administration lasted for 11 weeks, including the invasion period. After being supplemented with 2’-FL for 8 weeks, mice were challenged with 2 × 10^8^ CFU *E. coli* O157 (ATCC35150) every day, during which body weight was recorded, and stools were collected every other day. Mice were killed off one day after the last administration of *E. coli* O157, followed by blood and tissue collection. Blood serum was separated by centrifugation and stored at −80 °C until analysis. Two sections of ileum and colon, each 2 cm long, were immediately frozen in liquid nitrogen for RNA extraction. Cecal contents were taken and quickly frozen in liquid nitrogen for investigation of the gut microbiota. All animal protocols were approved by the Institutional Animal Care and Use Committee of Nankai University and performed in compliance with the national guidelines for experimental animal welfare and ethics. The strain used in this study was a low-virulence strain, which lacked *Stx2* (a virulence gene) compared to other *E. coli* O157 with high virulence. This weakened the damage in mice and reduced the number of deaths during long-term infection.

### 2.2. Determination of Infection in Mice

After the mice were sacrificed, 2 cm sections of ileum and colon were taken, opened lengthwise, and washed in sterile PBS. Intestinal tissues were homogenized using tissue homogenizer, from which bacterial DNA were extracted with a bacterial genome DNA extraction kit (TIANGEN BIOTECH, Beijing, China). The obtained bacterial genomic DNA was diluted to the same concentration using diethyl pyrocarbonate (DEPC)-treated water, and the relative content of *E. coli* O157 was determined by real-time fluorescent quantitative polymerase chain reaction. The primer sequences and methods for real-time fluorescent quantitative PCR (qPCR) can be found in Appendix A. The average colonization of *E. coli* O157 in the MC group was taken as the standard, and the colonization of each sample was calculated.

### 2.3. Preparation of Pathological Sections

Fresh intestinal tissue (0.5 cm) without feces was taken and placed in 2 mL tissue fixative solution, fixed at room temperature for 24 h, and then embedded in paraffin, stained with HE, and photographed for analysis.

### 2.4. Analysis of Inflammatory Levels in Serum and Intestine of the Mice

Inflammatory levels related to the nuclear factor kappa-B (NF-κB) pathway in blood, ileum, and colon (main sites of infection) were measured to characterize the representative inflammatory indicators of the host. Interleukin-6 (IL-6) and tumor necrosis factor-α (TNF-α) reflect overall acute inflammation, while interleukin-1β (IL-1β) represents the induction of inflammatory response. ELISA kits (Nanjing Jiancheng Bioengineering Institute, Nanjing, China) were used to detect contents of IL-6, TNF-α, and IL-1β in blood and intestine tissue. TRIzol (Invitrogen, Carlsbad, CA, USA) kits were used to extract RNA from tissue. cDNA synthesis was performed using the LunaScript SuperMix Kit (NEB, Ipswich, MA, USA). The expression of IL-6, TNF-α, and IL-1β in tissues was measured by qPCR (BIO-RAD, Hercules, CA, USA). The primer sequences and methods for qPCR can be found in Appendix A.

### 2.5. Analysis of mRNA Expression of Mucin- and Occludin-Related Genes

The expression levels of Mucoprotein-2 (*MUC2*) and occludin in the colon were determined by qPCR. The methods for extracting mRNA and qPCR are the same as in Section 2.2. The primer sequences can be found in Appendix A.

### 2.6. 16S rDNA Gene Sequencing of the Cecum of the Mice

Extraction of bacterial DNA from cecal contents, PCR amplicon sequencing, and gene analysis were all completed by Genedenovo Biotechnology Company (Guangzhou, China). 16S rDNA gene sequencing was adopted to analyze intestinal microbiome. The procedure included extracting microbial DNA and examining gene validity using gel electrophoresis. The specific primers with barcodes, 341F (CCTACGGGNGGCWGCAG) and 806R (GGACTACHVGGGTATCTAAT), were applied to amplify the V3 + V4 domain of 16S rDNA. The amplification was recovered and quantified using Qubit 3.0 fluorimeter, equal amounts of which were mixed and connected to the sequencing joint. A sequencing library was constructed according to Illumina official details and sequencing was accomplished using a PE250 model in Hiseq2500 system. After raw data were filtered and combined, operational taxonomic unit (OTU) results were obtained when 97% similarity was set as a crosscut. In order to reduce the influence of low-abundance OTUs on the whole analysis, OTUs with an abundance under 2 were filtered out without involvement in subsequent analyses. Based on the species richness information in the OTUs, the Chao1, ACE, Simpson, and Shannon indices were used to obtain estimated values of α-diversity. Arithmetic mean unweighted pair group method using arithmetic average (UPGMA) clustering, principal coordinate analysis (PCoA), and nonmetric multidimensional scaling (NMDS) were conducted to compare differences between intestinal bacterial community and forecast their function. The Tax4Fun algorithm was applied to prospect and analyze the function of the intestinal microbiome [30]. Tax4Fun first compared the prokaryotic 16S rRNA sequences of existing genomes in the KEGG database with 16S rRNA sequences in SILVA database, then the existing prokaryotic species genomes in the KEGG database were interrupted, and the KO sequences of all genomes were counted by UProC. Finally, the number of species was corrected by 16S copy number, and KEGG prediction and KO abundance statistics were finally implemented. Canonical correspondence analysis (CCA) and redundancy analysis (RDA) were employed to show the relationship between microbiota and environmental factor, then a distance matrix of species and environmental factors was calculated based on Bray-Curtis dissimilarity. A Mantel test was adopted to analyze environmental factors, and variable-precision arithmetic (VPA) was used to measure their influence on the microbial population.

### 2.7. Short-Chain Fatty Acid Quantification

A 0.2 g fecal sample was homogenized in 1 mL pure water and the solutions were centrifuged (10,000× *g*, 5 min) to obtain supernatant fluid. After being acidulated using sulfuric acid, short-chain fatty acids (SCFAs) were then extracted using a 0.5 mL ether. The mixture was well mixed by 2 min vortex agitation and then left to stand for 10 min. The organic ether phase was then removed carefully from the lower strata and filtered through a 0.45 μm nylon membrane, which was then placed in brown autosampler vials for later use. DB-FFAP capillary column (30 m × 0.25 mm × 0.25 μm, Agilent) was used for gas chromatographic analysis. Agilent’s MSD ChemStation (E.02.00.493) was adopted for data processing.

### 2.8. Cell Culture and Adhesion Experiments

Caco-2 cells are a kind of clonal human colonic adenocarcinoma cell whose structure and function are similar to differentiated intestinal epithelial cells [31]. DMED cell culture medium (Beyotime, Shanghai, China) was used for culturing Caco-2 cells in 75 cm^2^ culture flasks under 37 °C with 5% CO_2_ and humidified air. Our culture medium also contained fetal calf serum, 1% nonessential amino acids, 1% glutamine, and Penicillin-Streptomycin Solution (100×) (Beyotime, Shanghai, China) to ensure sterility. When the cell monolayers covered 90% area of the culture vessel, the cells were then trypsinized (Beyotime, Shanghai, China) and counted. A 1 mL suspension with 6 × 10^5^ cells was added into each well of a 12-well tissue culture plate, and half of the culture medium was renewed every 2 days. After culturing for 14 days, previous culture medium was substituted with culture medium without antibiotic. Cells could then be used for experiments after culturing for 24 h.

2’-FL and D-lactose were added into the cell suspensions to final concentrations of 0.5, 2, 5 g/L, among which D-lactose served as a non-inhibiting control. A blank control was set at the same time. A total of 1 × 10^8^ CFU/mL *E. coli* O157 mixed with cell culture medium was inoculated into each well, and the whole invasion process lasted for 1 h in order to prevent excessive proliferation of bacteria and damage to cells. After that, the cells were washed by preheated Hanks’ balanced salt solution to dislodge non-adherent bacteria, and then 0.025% Triton X-100 (Sigma Aldrich) was used to release bacteria that were adherent to cells without affecting their viability. Dilution counting was applied to quantify the number of bacteria that adhered to cells in each well. The number of bacteria in wells without glycan was taken as the total number, and the adhesion in other wells was calculated. All experiments were repeated three times, with three parallel experiments for each condition. The survival of cells after infection by bacteria was determined using an alamarBlue cell viability assay (Beijing Solarbio Science & Technology Co., Ltd., Beijing, China) following the given instructions, whereby the final results were obtained using ultraviolet spectrophotometry.

### 2.9. Statistical Analysis

The difference between two groups was analyzed using the two-tailed Student’s *t*-test. The difference between more than two groups was analyzed using one-way ANOVA followed by a Duncan post hoc test. All the data were rendered into a mean ± standard deviation. One-way ANOVA or independent *t*-tests were conducted using GraphPad Prism 8.0.2 (GraphPad Software, San Diego, CA, USA). When *p* < 0.05, differences among groups of data obtained from independent trials were considered statistically significant. Bioinformatic analysis was performed using Omicsmart, a dynamic real-time interactive online platform for data analysis (http://www.omicsmart.com).

## 3. Results

### 3.1. 2’-FL inhibit Colonization of E. Coli O157 in Intestine

The characteristic virulence gene *Stx1* was used to evaluate the colonization of *E. coli* O157 in the intestine [32]. As shown in Figure 1, in ileum (Figure 1A), there were few *E. coli* O157 in the CK group, and only 8.63% ± 6.87% in the FL group, so a highly significant difference can be seen with the MC group, which demonstrates that intake of 2’-FL greatly reduced *E. coli* O157 colonization in the ileum. In the colon (Figure 1B), barely any *E. coli* O157 colonization can be found in the CK group, and the low amplification can be regarded as a trace amount compared to the MC group, and may have resulted from other microbes in cecum which have the *Stx1* virulence gene. The colonization rate was only 6.16% ± 2.25% in the FL group, which presented a highly significant difference with the MC group. As a type of oligosaccharide, 2’-FL has similar results to those observed for many other oligosaccharides, but the specific effects on microbiota and the intestinal environment need further investigation [23].

An ileal tissue slice to examine pathology revealed that although some part of infiltration can be seen in the crypt of villus in the FL group (Figure 1D), the villus was healthy in general, with a neat and compact construction. On the other hand, the villi of the MC group (Figure 1E) were greatly damaged, crypt cells were infiltrated markedly, and inflammatory cells were concentrative and activated. All of these findings indicate that 2’-FL protects intestinal health in mice and reduces the incidence of pathological changes at a pathological level.

### 3.2. 2’-FL Lowers Inflammatory Levels and the mRNA Expression of Mucin- and Occludin-Related Genes

Following the instructions of ELISA kits, cytokines were measured using ELISA and the results are shown in Table 1. Regardless of whether it was in ileum, colon, or serum, inflammatory levels were significantly lower in the FL group than in the MC group. Among these, cytokine content in the colon was the lowest, then the serum, and the highest appeared in the ileum; this might be because ileum was the main lesion site of *E. coli* O157.

A similar tendency was reflected in the mRNA expression of inflammatory cytokine-related genes (Figure 2). Compared with the CK group, the levels of TNF-α, IL-6, and IL-1β all soared under infection with *E. coli* O157, and the FL group all decreased to normal levels with the intervention of 2’-FL, with no statistically significant differences observed for the CK group. Mucin gene expression results displayed that *MUC2* and occludin expression in the MC group were markedly lower than those of the other groups, and that of the FL group was much higher than other groups. This shows that 2’-FL can effectively promote the secretion of intestinal mucin and occludin.

### 3.3. Effects of 2’-FL on Gut Microbiota Composition

In order to explore the effect of 2’-FL on the intestinal microbiome in a mice infection model, samples of stools were taken for sequencing and corresponding bioinformatics analysis. The results are shown in Figure 3. At the phylum level (Figure 3A), Firmicutes, Bacteroidetes, and Proteobacteria were the dominant bacteria in the MC group. Compared with the CK group, there was an obvious reduction in Deferribacteres and Patescibacteria; however, Proteobacteria and Verrucomicrobia were found to be enriched, which is in line with the results of most studies. As *E. coli* O157 belongs to Proteobacteria, it was quite easy to understand the increase in Proteobacteria. After the long-term supplementation of 2’-FL, compared with the CK group, Firmicutes and Patescibacteria were sharply reduced in response to large doses of *E. coli* O157, and there was a striking difference in Verrucomicrobia between the CK and MC group (*p* < 0.001). The proportion in the CK group was 1.76%, and it was 6.34% in the MC group, while in the FL group, it was 26.42%. Additionally, Firmicutes was upregulated in the FL group. At the genus level (Figure 3B), *Parabacteroides*, *Eubacterium*, *Mucispirillum*, *Anaerotruncus*, and *Alistipes* were significantly reduced in the MC group (*p* < 0.05), and *Candidatus_Saccharimonas*, *Lachnospiraceae_NK4A136*, and *Alistipes* in the FL group were downregulated (*p* < 0.05). In the FL group, *Lactobacillus* and *Lachnospiraceae_NK4A136* were significantly lower than in the MC group (*p* < 0.05), but *Ruminococcaceae* and *Akkermansia* were higher (*p* < 0.05 and *p* < 0.001).

A dilution curve can be used to directly quantify species, and the height of the curve indirectly reflects the abundance of species. As shown in Figure 3C, species abundance in the CK group was similar to the MC group, which is in accordance with previous studies. However, that of the FL group was lower than for the other groups. Using PCoA analysis (Figure 3D), clusters between groups are obvious, among which species diversity in the MC g and CK groups were smaller. Consulting the SILVA database, in Level 2, functions of folding, sorting, and degradation declined in the MC group compared with the CK group, while the expression of pathways associated with cancer increased significantly (*p* < 0.01) according to the functional gene forecast. In the FL group, the endocrine system and digestive system abilities were noticeably weakened, while ABC transporters, arginine and proline metabolism, and nitrogen metabolism ability improved significantly (*p* < 0.05). The nitrogen metabolism ability was enhanced in the MC group compared to the CK group in Level 3 (Figure 3E) (*p* < 0.05). The FL group had a better performance in terms of methane metabolism; starch and sucrose metabolism; alanine, aspartate, and glutamate metabolism; amino sugar and nucleotide sugar metabolism; cell cycle—*Caulobacter*; fructose and mannose metabolism; and purine metabolism compared to the MC group.

To research the correlation between gut microbiota and other sample indicators, we introduced an environment factors analysis and built a CCA model for microbiota samples and five main factors, including colon infection rates, overall amount of SCFAs, *MUC2* expression and IL-6 and TNF-α expression in the colon. As shown in Figure 4A, CCA1 and CCA2 accounted for 61.58% and 30.11% of the whole microbiota variation, respectively; and the correlation, significance, and contribution of different factors are listed in Figure 4B. The result showed that *MUC2* expression and the infection rate were highly significantly correlated with the distribution of gut microbiota. As seen from the correlations among species, Figure 4C shows that there was a significant positive correlation between *Akkermansia* and *MUC2* expression (*p* < 0.001), and between *Escherichia* and infection rates and the presence of IL-6 and TNF-α (*p* < 0.01). *Lachnoclostridium* had similar results to *Escherichia*, and a significant negative correlation with SCFAs and *MUC2* was also observed (*p* < 0.05).

### 3.4. 2’-FL Increases Content of SCFAs in Colon

The contents of acetic acid, propanoic acid, butyric acid, and pentanoic acid were determined, and the results are shown in Table 2. The overall amount of SCFAs in the FL group was 66.76% ± 19.63% higher than the MC group (Figure 5A), which is highly significant (*p* < 0.001). Each type of SCFA in the experimental group is also significantly higher than the MC group (*p* < 0.01). In the FL group, the average level of acetic acid showed a 52.94% ± 21.06% increase and butyric acid increased by 89.92% ± 21.76% compared with the MC group. It is noteworthy that the average level of propanoic acid displayed a 106.7% ± 29.63% increase. It is uncommon to see such a sharp increase of more than double for a certain type of SCFA, which implies wide variations in either the intestinal environment or microbiota.

As most prebiotics can effectively act on increasing SCFAs in intestine, a single evaluation of 2’-FL protection by measuring differences in SCFAs after infection does not represent the complete story. The SCFA levels before infection and after being supplemented with 2’-FL were also determined, and similar results were attained, showing that ingesting 2’-FL raised the level of SCFAs in the colon (detailed data can be seen in Appendix A). After comparing data before and after infection (Figure 5B), it was obvious that *E. coli* O157 significantly lowered the level of SCFAs in mice (*p* < 0.001), but the extent of reduction in the FL group was much less than in the MC group. This demonstrated that a generous intake of *E. coli* O157 has a huge impact on the intestinal environment, and 2’-FL taken orally in advance may defend against this impact.

### 3.5. 2’-FL Inhibits Adhesion of E. Coli O157 to Cells

Cell adhesion experiments (Figure 6A) made it clear that 2’-FL can dramatically decrease *E. coli* O157 adhesion on Caco-2 cells. If the *E. coli* O157 adhesion rate of the MC group was considered as 100%, the adhesion of low-concentration in the FL group can be counted as 97.73% ± 4.06%, not differing statistically with the MC group (*p* = 0.39). Adhesion of the medium-concentration FL group was 79.89% ± 7.99%, while it was only 65.99% ± 12.35% in the high-concentration group, so significant differences were detected. A negative correlation between *E. coli* O157 adhesion number and 2’-FL concentration was observed (r = 0.97, *p* = 0.08), while there was no significant difference between lactose of different concentration groups and the MC group. All of these findings illustrate that 2’-FL did play a role in reducing the adhesion of *E. coli* O157 on Caco-2 cells and the effect increased progressively with increased concentrations.

Cell vitality was determined following manufacturer’s instructions (alamarBlue, Beijing Solarbio Science & Technology Co., Ltd.) and results were similar to adhesion results (Figure 6B). When cells were supplemented with 5 g/L 2’-FL, the vitality was significantly higher than group with lactose intervention or the MC group without any intervention (*p* < 0.01). This demonstrates that certain concentrations of 2’-FL have a protective effect on cells against *E. coli* O157 infection.

## 4. Discussion

Clinical studies have shown that HMOs, especially 2’-FL, can reduce infant diarrhea and gastrointestinal infection in humans [33,34]. They also had an effect on enhancing rat immunity functions, relieving colitis severity, inhibiting *Campylobacter jejuni* colonization, and reducing *Rotavirus*-induced diarrhea in rodents [35,36]. However, the protective effect of long-term intake of 2’-FL and its regulation of intestinal microbiome, as well as the potential mechanisms involved, are yet to be established. In the present study, a three-month intervention of 2’-FL significantly reduced the colonization of *E. coli* O157 in both the large and small intestine. Among them, the colonization rate of the ileum and colon decreased by 91.4% and 93.8%, respectively. The administration of 2’-FL also suppressed the inflammatory response in the intestine and improved the integrity of intestinal villi. Compared with the model group, 2’-FL reduced IL-6 by 39.2%, IL-1β by 53.0%, and TNF-α by 37.4%. Moreover, we found that the preventive effect of 2’-FL against harmful microorganism infection is not only related to its competitive binding of specific oligosaccharide structure, but also largely depended on its specific modulation of the gut microbiota and gut barrier.

The first step of pathogenic bacteria infection is to adhere and colonize the intestinal tract, and the number of adhesions directly reflects the infection situation [37]. Our animal experimental results indicate that 90% of *E. coli* O157 colonization in the intestine was reduced by 2’-FL; in other words, 2’-FL greatly alleviated *E. coli* O157 infection. This protective effect is similar to observation in previous research [17]. Ruiz-Palacios et al. found that α1,2-linked fucose can inhibit *Campylobacter* from binding with H-2 antigen in the intestine, so its binding can be inhibited by 2’-FL, which contains this structure. It is considered that the alimentary canal of humans and mice cannot secrete specific types of galactosidase and 2’-FL cannot be digested by the stomach, so most remains in its original structure when reaching the intestine. Earlier studies found that HMOs, having a similar structure to cell surface receptors, act as a soluble decoy receptor and greatly reduced germ colonization on the intestinal epithelium, thereby lowering the incidence of gastrointestinal infectious disease [14]. Here, we would like to express our heartfelt thanks to the scientific researchers who conducted research in the mechanism earlier. Their work was very inspiring for the design and discussion of this experiment. In the present study, the co-culture of 2’-FL and Caco-2 cells significantly reduced the amount of *E. coli* O157 adhesion on cell surface, and the proportion of reduction increased with the concentration of 2’-FL. Among them, the amount of adhesion decreased by 34% at the concentration of 5 g/L. Our cell experiment data suggest that 2’-FL can reduce the adhesion of *E. coli* O157 on the cell surface and the effect was positively correlated with the concentration of 2’-FL. It has been shown that 2’-FL could cut down the colonization of several pathogens on the cell surface, including *Campylobacter* spp., enteropathogenic *Salmonella* spp., and *Pseudomonas* spp., which is consistent with our results. In sum, our research found that the protective effects of 2’-FL were mediated by reduction in the in vivo adherence of harmful microorganisms.

Based on the results of bacterial sequencing, we consider that 2’-FL has a powerful capacity to intervene with the host intestinal microbiome, which is mainly embodied in the great differences in reforming the host microbiota and the stability of these differences. From the PCoA analysis (Figure 3D), species differences between the MC and CK groups were smaller, which indicates that taking large oral doses of *E. coli* O157 had less influence on bacterial diversity in mice than 2’-FL. From the 16S sequencing results, we can see that in the FL group, species abundances increased in Parabacteroides, which has the function of balancing blood fat; in Ruminococcaceae, which controls production of SCFAs; and in *Akkermansia*, which can stimulate mucin production in hosts. In the MC group, species richness decreased for such opportunistic pathogens as *Sphingomonas*, *Luteibacter*, *Mycobacterium*, as well as *Actinomadura*, which can promote inflammation. All these findings point to the tremendous changes in microbiota under intervention by 2’-FL, whose microorganisms were also healthier when infected by *E. coli* O157. It was more intuitively seen that when mice are given large amounts of *E. coli* O157 orally, the abundance of *E. coli* significantly increased, but that of the FL group was significantly lower than the MC group, even though a an extremely high dose of *E. coli* O157 (2 × 10^8^ CFU) was used for 3 weeks in our experiments. Furthermore, we also found that species belonging to genus *Lachnospiraceae_NK4A136* were dramatically reduced after intervention with 2’-FL, but in previous studies, it was considered that the abundance of *Lachnospiraceae_NK4A136* was positively correlated with *Ruminiclostridium_6* and *Ruminococcaceae_UCG-014*, and all are desirable probiotics that can improve digestion and produce SCFAs [38,39]. We suspected that the abundance of *Ruminiclostridium_6* and *Ruminococcaceae_UCG-014* increased under the influence of 2’-FL, and when their communal nutrients are fixed, certain competitive relationships are formed; therefore, the abundance of Lachnospiraceae declined. Overall, the abundance of *Lachnospiraceae_NK4A136* in the MC group was still 10% higher than in the FL group, and still played a dominant role at the genus level. Interestingly, we unexpectedly observed an increase in the *Lactobacillus* in MC group and this is similar to previous studies by Wang, W. et al. [40]. This may be due to a reduction in the abundance of some strains in the intestinal tract of infection models, whereas other strains like *Lactobacillus* have a space-occupying effect. Nevertheless, the abundance of Lactobacillus was still relatively low (all below 2%) in all groups in the experiment. However, this is only a hypothesis and we will continue to focus on it in future studies. Remarkably, examining the heatmap of environment factors, *Lachnoclostridium*, a representative biomarker of harmful bacteria, was positively correlated with both the expression of inflammatory cytokines and infection rates, while it was negatively correlated with SCFAs and mucin expression in our experiments. Few studies have addressed species of this genus, especially its function in gut health, and have only concerned the decomposition of lignin and metabolism of tryptophan [41,42]. *Alistipes*, *Ruminiclostridium*, *Bilophila*, *Mucispirillum*, and *Eubacterium xylanophilum* were similarly closely associated with environmental factors, so concerns need to be raised regarding them. In brief, our research findings show that 2’-FL intervention can reduce the abundance of harmful microorganisms in mice intestine, and greatly modify the profiles of intestinal microorganisms in mice towards a healthier and more stable direction.

SCFAs have always been considered beneficial to human health by directly providing energy for intestinal epithelial cells and liver, and their function in reducing intestinal pH also helps to resist invasion by harmful bacteria [43]. Through determination of SCFAs in mice stools, gavage administration of 2’-FL can significantly improve SCFA content in the intestine, which is consistent with the results of previous studies [44,45]. These SCFAs were from two sources. One was produced by microbiota utilizing 2’-FL in intestinal tracts. Zabel et al. fermented 2’-FL using *Bifidobacterium* spp. and determined the production through track mensuration [24]. They discovered that 2’-FL can be degraded by *Bifidobacterium* spp., and that, in this process, acetate and propionate were produced [46]. The other came from the increased abundance of species which produce SCFAs after 2’-FL supplementation, such as *Akkermansia* and Ruminococcaceae. Interestingly, it is usually considered that the phyla Bacteroidetes and Firmicutes are the dominant microorganisms producing SCFAs, but the abundance of Firmicutes in the FL group was significantly lower than in the CK or MC groups in our experiment, while Bacteroidetes were less abundant than the CK group and there was no difference with the MC group. The FL group had a significantly higher SCFA content than the MC group. These findings were distinct from the traditional sense of the relationship between Bacteroidetes and Firmicutes abundance and SCFAs [47,48]. We ascertained that the metabolic processes that produce SCFAs under intestinal conditions are quite complex, and the process of a species generating SCFAs under specific conditions needs to be described more accurately, instead of simply overgeneralizing from a large taxon unit. Moreover, SCFAs can play a role as HDAC inhibitors, which limits downstream NF-kB signal transduction to a certain degree, thereby benefiting intestine health [49]. In our data, both the contents and expression of inflammatory cytokines in the FL group were significantly lower than the MC group; this was partly due to the intake of 2’-FL in suppressing colonization of *E. coli* O157 and relieving inflammation, and partly because SCFA can block HDAC and silence the NF-kB pathway. In summary, like for most other prebiotics, 2’-FL can significantly increase the level of intestinal SCFAs and promote gut health from a chemical standpoint.

Gut barrier function is a key indicator of gut health, as well as the major line of defense against external infection. As an important physical barrier, the mucus layer can physically restrain the colonization of pathogens on the enterocyte [50]. Earlier studies have revealed that goblet cells are specialized in the secretion of mucin and formation of the mucus layer [51]. As mucin is produced from inside out and due to the motility of intestinal villi, uncolonized bacteria were eliminated from the body quickly. Consistent with this, in our research, the expression of *MUC2* and occludin were observed to be upregulated in the FL group. A previous study also suggested that SCFAs contribute to increased mucin expression in goblet cells, and we have shed light on the decomposition effect of *Akkermansia* spp. on mucin. All of these findings reveal a subtle interplay: *Akkermansia* spp. resolves mucin as its own source of carbon and nitrogen and produces SCFAs while stimulating goblet cells to secrete more mucin at the same time. A balance between *Akkermansia* spp. abundance, SCFA level, and mucin content is formed in the intestine, namely, the Akkermansia-SCFA-mucin balance (A–S–M balance). This balance is stable within a certain range, where a minor change of one would not bring an immediate change to the other two; however, when such great change takes place with one of the contents that the balance is upset, the body’s internal environment would adjust the other two until a new balance is formed. For example, many animal experiments have found that the slime layer was destroyed and/or mucin expression was blocked in diseases such as inflammatory bowel disease (IBD) and irritable bowel syndrome (IBS), during which all three elements decreased synchronously and the A–S–M balance ultimately became low-level [52,53,54]. Similar consequences were found in research involving dietary fiber and resistant starch, as substantial growth in the level of SCFAs, mucin content, and *Akkermansia* spp. abundance were observed [43,55,56,57]. Studies have indicated that *Akkermansia* spp. proliferates by utilizing 2’-FL, which is consistent with our results [58]. A 15-fold increase and 26.4% proportion at genus level showed the absolute predominance of *Akkermansia* in the FL group compared with the CK group. More SCFAs being produced and more mucin being discharged could be interpreted as the mucin layer becoming more substantial and solid, and the A–S–M balance becoming higher level. According to our analysis of environmental factors, *Akkermansia* spp. was the only type of bacteria that presented a significant positive correlation with *MUC2* expression, which further evidences the existence of an A–S–M balance. Hence, our study filled an important gap for the A–S–M balance hypothesis. The stimulation of mucin secretion by *Akkermansia* spp. was also mentioned by other researchers, explaining that other signal mechanisms may exist, except for those depending on SCFAs [58]. This is not in contradiction with our results because A–S–M balance is not an enclosed system; on the contrary, it is open to other mechanisms between A and M. In fact, these constitute an A–Xs–M balance, which we will not address further here. However, although no abnormal changes were observed in our experiment, the explosion in *Akkermansia* spp. proliferation and whether, after cessation of 2’-FL supplementation, the overwhelming amount of *Akkermansia* spp. would lead to adverse effects still requires further investigation. In conclusion, we believe that the protective effect of 2’-FL is mediated by the specific proliferation of *Akkermansia* spp. (Figure 7).

Our experimental results only used an adult mouse model. Taking the great differences between gut microbiota in mice and humans into consideration, further research is needed. For example, the abundance of *Bifidobacteria* spp. in the adult gut—A beneficial bacteria widely reported to be able to promote growth by 2’-FL [25]—Is reported at around 5%, whereas in this study, almost no *Bifidobacteria* spp. were detected in adult mice. Moreover, more extensive research is required to know whether the modulating effect of 2’-FL on gut microbiota correlates with its level of intake. Whether an increase or decrease in intake can still specifically promote proliferation of *Akkermansia* spp., which plays a protective role, is also an issue that needs to be explored. In addition, the strategy adopted in this experiment is a long-term and low-toxicity *E. coli* O157 infection strategy, and the hosts’ responses were different in the face of different virulence intensity, dose and duration and other intervention conditions. The discussion of protective mechanism of 2’-FL in this study is currently limited to this experimental scheme. A more realistic model should be constructed in the future for more universal mechanism research, for example, the use of more common strains or animal models that are closer to the human gut situation, and the use of infection strategies that are closer to the true course of infection.

## 5. Conclusions

In conclusion, we demonstrated that 2’-FL can inhibit the colonization of harmful bacteria through cell surface receptor binding and the enhancement of intestine barrier function. We firstly found that 2’-FL could selectively increase the abundance of *Akkermansia* spp. and upregulate the expression of mucin related to gut barrier in mice. Recent studies have shown that *Akkermansia* spp. may be associated with the treatment of certain chronic diseases, such as obesity and diabetes, so 2’-FL selectively increasing the abundance of *Akkermansia* spp. may provide new ideas for the treatment and prevention of these diseases. In summary, our study sheds light on the protection conferred by 2’-FLs and the mechanism behind its defense against pathogen infection, and offers a basis for learning about the modulation of 2’-FL on human gut health. With the gradual industrialization of the production of HMOs such as 2’-FL, more research should be conducted on intervention with 2’-FL or other types of HMOs on the population or animal models other than infants, so as to promote the development of human beings in a healthier direction.

## Figures and Tables

**Figure 1 nutrients-12-01284-f001:**
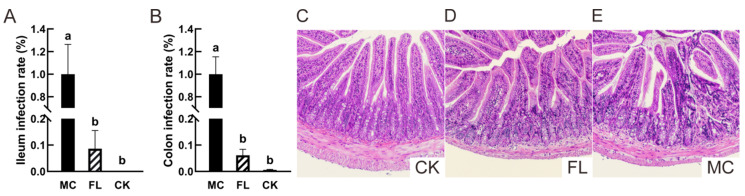
*Escherichia coli* O157 colonization and pathological section of ileum. (**A**) The colonization in ileum; (**B**) colonization in colon; (**C**) representative images of ileum tissue sections in the blank (CK) group (HE 100×); (**D**) representative images of ileum tissue sections in the FL group (HE 100×); (**E**) representative images of stained ileum tissue sections in the mode control (MC) group (HE 100×). Different letters (a, b) indicate significant differences among groups (*p* < 0.05).

**Figure 2 nutrients-12-01284-f002:**
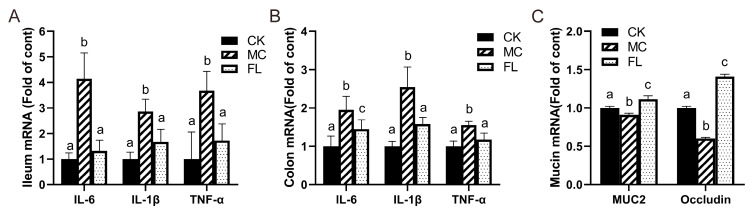
Relative mRNA levels of inflammatory cytokines and mucins, all normalized to β-actin mRNA expression. (**A**) Inflammatory cytokines in ileum; (**B**) inflammatory cytokines in colon; (**C**) expression of mucin in colon. Different letters (a–c) indicate significant differences of the same factor among groups (*p* < 0.05).

**Figure 3 nutrients-12-01284-f003:**
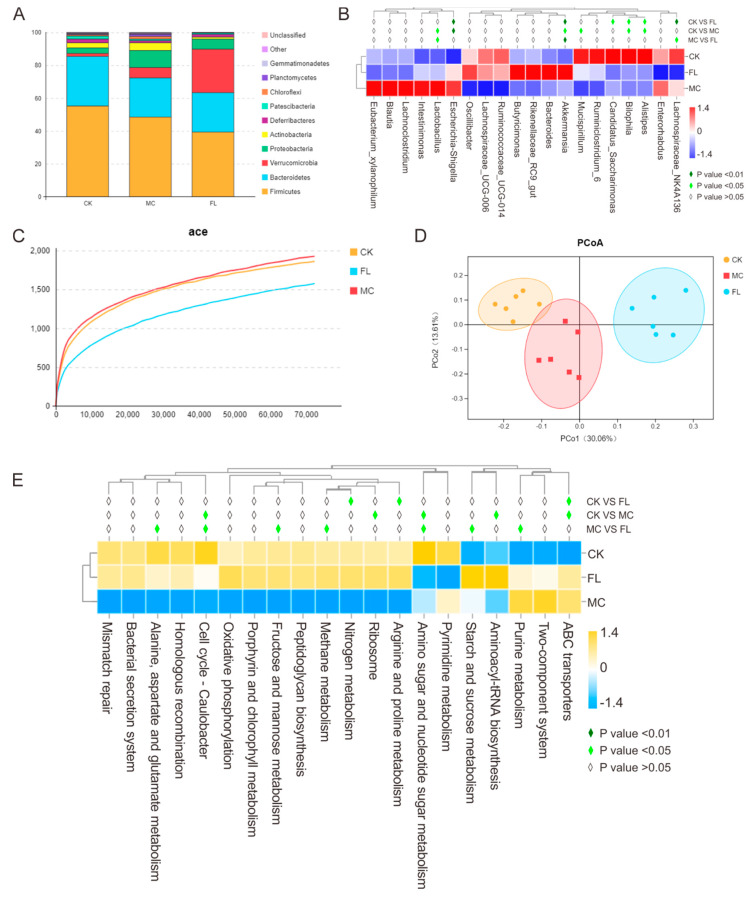
Relative abundance, diversity, and function prediction of intestinal microbiome in different groups of mice: (**A**) Phylum level; (**B**) genus level; (**C**) dilution curve; (**D**) PCoA analysis based on operation taxonomic unit (OTU) results; (**E**) Tax4Fun analysis based on SILVA database.

**Figure 4 nutrients-12-01284-f004:**
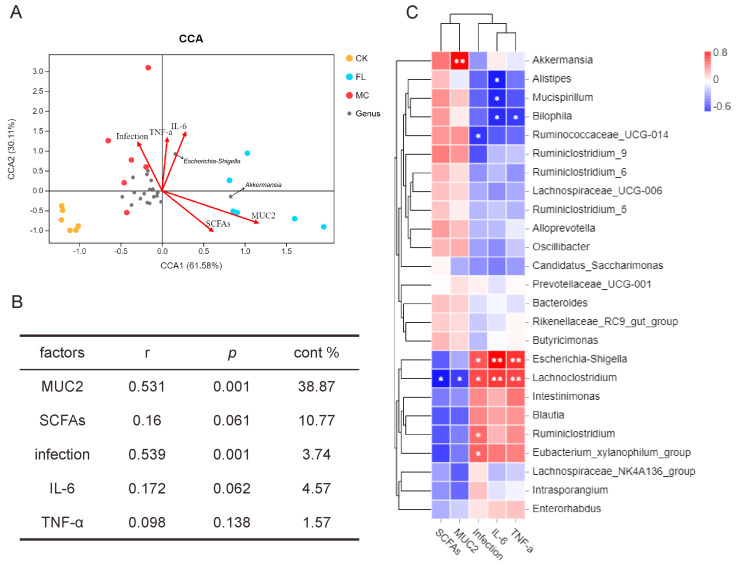
Correlation analysis of environmental factors. (**A**) Canonical correspondence analysis (CCA) analysis of microbial population distribution and environmental factors. Environmental factors are marked by arrows. The angle between the arrow line and the sorting axis indicates the correlation between environmental factors and the sorting axis. Environmental factors with a longer arrow had a higher correlation with the distribution of intestinal microbiome. (**B**) The significance of environmental factors was analyzed by envfit function. r is the determining coefficient of environmental factors on bacterial distribution; the larger the r value, the greater influence this environmental factor has on microbiome distribution. *p* is the significance level of correlation test. (**C**) As shown in the Spearman correlation thermogram, there were significant differences in the abundance of bacteria between different environmental factors. The colors range from blue (negative correlation) to red (positive correlation). A significant correlation is indicated by * *p* < 0.05 and ** *p* < 0.01.

**Figure 5 nutrients-12-01284-f005:**
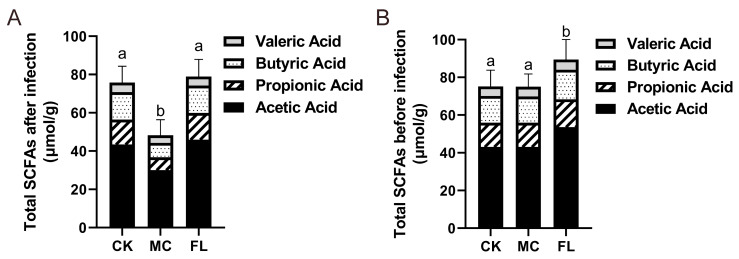
Total amount of short-chain fatty acids (SCFAs) in feces before and after infection. (**A**) Before infection; (**B**) after infection. Levels of different types of SCFAs are reflected. Different letters (a, b) are used to express significant differences between total SCFAs (*p* < 0.05).

**Figure 6 nutrients-12-01284-f006:**
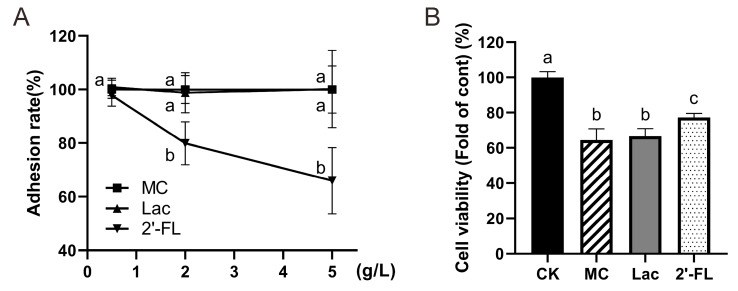
Adhesion rate of *E. coli* O157 and cell viability under infection. (**A**) Adhesion rates in each group at different concentrations; (**B**) cell viability. Significant differences are indicated by different letters (a–c) (*p* < 0.05).

**Figure 7 nutrients-12-01284-f007:**
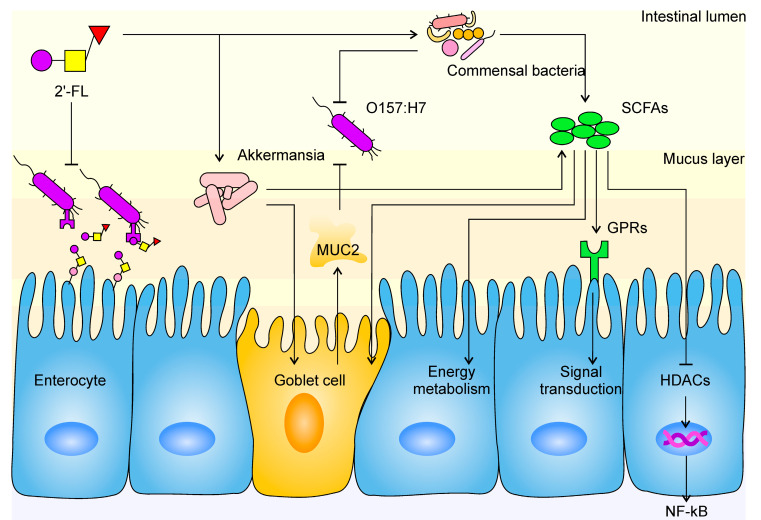
The mechanism of 2’-FL protection against *E. coli* O157 infection in the intestine. The tapered arrow indicates a promoting effect, while the flat end indicates an inhibiting effect. Mucoprotein-2, *MUC2*; short-chain fatty acids, SCFAs; histone deacetylases, HDACs; G protein-coupled receptors, GPRs; nuclear factor kappa-B, NF-κB.

**Table 1 nutrients-12-01284-t001:** Concentration of inflammatory cytokines in different tissues from different groups.

	Group	IL-6 (pg/mL)	IL-1β (pg/mL)	TNF-α (pg/mL)
Serum	CK	76.61 ± 6.02 ^a^	13.86 ± 1.45 ^a^	35.78 ± 6.19 ^a^
	MC	165.87 ± 5.63 ^b^	37.42 ± 2.42 ^b^	78.24 ± 5.75 ^b^
	FL	100.85 ± 8.44 ^c^	17.59 ± 1.21 ^c^	48.99 ± 5.40 ^c^
Ileum	CK	170.72 ± 13.00 ^a^	16.82 ± 0.97 ^a^	40.34 ± 3.19 ^a^
	MC	210.68 ± 8.97 ^b^	42.03 ± 2.49 ^b^	77.89 ± 6.49 ^b^
	FL	191.19 ± 9.66 ^c^	28.67 ± 1.72 ^c^	49.26 ± 4.06 ^c^
Colon	CK	74.95 ± 3.77 ^a^	14.64 ± 0.74 ^a^	36.77 ± 1.85 ^a^
	MC	160.40 ± 8.06 ^b^	34.30 ± 1.72 ^b^	60.14 ± 3.03 ^b^
	FL	147.38 ± 7.40 ^a^	25.62 ± 1.29 ^c^	45.61 ± 2.29 ^c^

Data are expressed as mean ± SD (n = 8). Values in the same column labeled with different superscripts (a–c) differ significantly (*p* < 0.05). Interleukin-6, IL-6; tumor necrosis factor-α, TNF-α; iInterleukin-1β, IL-1β.

**Table 2 nutrients-12-01284-t002:** Concentration of SCFAs in the feces of each group.

Group	Acetic Acid (μmol/g)	Propionic Acid (μmol/g)	Butyric Acid (μmol/g)	Valeric Acid (μmol/g)
CK	43.45 ± 5.92 ^a^	12.98 ± 2.37 ^a^	14.32 ± 1.90 ^a^	5.04 ± 0.73 ^a^
MC	30.06 ± 6.97 ^b^	6.74 ± 1.68 ^b^	7.51 ± 1.58 ^b^	3.92 ± 0.41 ^b^
FL	45.98 ± 4.97 ^a^	13.93 ± 1.90 ^a^	14.26 ± 1.71 ^a^	4.77 ± 0.30 ^a^

Data are expressed as mean ± SD (n = 8). Values in the same column labeled with different superscripts (a, b) differ significantly (*p* < 0.05).

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
