# Peer review of "The Protective Effects of 2’-Fucosyllactose Against E. Coli O157 Infection Are Mediated by the Regulation of Gut Microbiota and the Inhibition of Pathogen Adhesion"

_nutrients, 2020, doi:10.3390/nu12051284_

Round 1

Reviewer 1 Report

The manuscript “Protective effects of 2’-FL against E. coli O157 infection is mediated by the regulation of gut microbiota and the inhibition of pathogen adhesion” describes interesting work on the beneficial effects of consumption of 2’-FL and the causative mechanisms associated with these effects using mouse model, molecular, metagenomic and microscopic methods. Statistical methods have been applied to analyse and interpret the data. The manuscript has been written well and the following are my concerns, suggestions and comments for authors’ consideration.

First, the authors must acknowledge earlier studies that have clearly demonstrated the antiinfection (especially E coli and C jejuni) effects of 2-FL both in vitro and animal model systems and discuss their observation in context of what is already known. Especially, studies by Elison and colleagues (BJN, 2016) on E. coli new-born piglet model. Authors need to know that new-born piglet model is the best animal system, compared with mouse models, for examining responses to human gut pathogens. Also, I have serious concerns about the quality/appropriateness of the E coli O517 mouse model described by the authors. The animal experimental design does not give relevant details, eg. Experimental group sizes, acclimatisation period, and treatment period. The method states that the 6-week old mice were challenged with a large dose of 2×108 CFU E. coli O157 (ATCC35150)/mouse for 3 weeks. The concern is, how could the mice survive such a large dose for a prolonged period. In the previous infection models the challenge dose is once (Wadolkowski et al. 1990) (1010 cfu/mouse) and the disease symptoms were seen in one week. In this study, no mortalities were reported in MC group. Another question that arises is whether the E coli O157 used is a non-virulent (?) strain. It may be speculated that the challenge organisms can utilise mucin, proliferate and affect the populations of Firmicutes and Bacteroidetes. It also raises ethical concerns of using overdose of pathogenic strain in mice. The authors need to provide a strong argument to support that the challenge method used is valid.

Minor suggestions and comments:

  1. Please italicise ‘in vitro’ and ‘in vivo’ wherever used in the text. Also there is no need to italicise ‘Enteropathogenic’ wherever used in the text.
  2. Line 149, please make correction to reflect ‘6x105
  3. Line 152, please change ‘cultured’ to ‘culturing’
  4. Line 155, please insert ‘medium’ after ‘culture’
  5. Line 160, ‘coating’ or ‘counting’? clarify
  6. Fig 3B – please discuss why Lactobacillus abundance increased along with Escherichia-Shigella in MC
  7. Line 334 – change .17 to (17).
  8. Line 335 – change ‘Guillermo M. Ruiz-Palacios et al’ to Ruiz-Palacios and co-workers
  9. Line 345 – insert ‘was’ after ‘effect’
  10. Line 346 – insert ‘the’ after ‘with’
  11. Line 350 While discussing microflora changes, it would be nice to discuss briefly the difference between mouse and infant microflora
  12. Line 385 – use of definite words ‘absolutely’ may not be accurate in discussing results. Please change the word to eg. ‘significantly’
  13. Line 387 – Zabel, Bryan et al – reference is missing
  14. Line 404 – ‘In a word’ – replace with ‘In summary’
  15. Line 426 – ‘in line with our promotion’ – please use more appropriate terminology
  16. Line 435 – please change 54 t0 (54)
  17. Line 447 – ‘competitive combination’ – does not make sense – did you mean ‘cell surface receptor binding’
  18. Line 452 – ‘In a word’ – replace with ‘In summary’
  19. Line 457-458 – the sentence does not make accurate sense, please modify
  20. Line 614 – please provide complete reference (PhD thesis)

Author Response

请参见附件。

Reviewer 2 Report

Dear Sirs,

Authors performed a mice study and cell culture experiments to investigate an effect of oligosaccharides on infection of enterohemorrhagic E.coli O157:H7. In the mice model they studied the colonization/ adhesion of this pathogen, its effect on peripheral and gut inflammation, as well as the expression of mucins. They also investigated the content of fecal short chain fatty acids and the fecal microbiome content after infection. The authors have performed a series of fine experiments and got certain results but those are is poorly presented. Unfortunately, the whole manuscript is unprecise, p-values are no or fairly presented, and terminology is lost; major revision is needed. This is a draft not a manuscript. Authors should use a consultation of microbiologist / molecule microbiologist for the details and corrections. The English language should also be checked. The draft is difficult to evaluate due to many shortages. Please see my comments below.

Major comments

  1. in the title: please do not use abbreviations like 2'-FL at the first time in the title. Change the title or use full name.
  2. In the abstract please identify the aims clearly and add how the goals were achieved e a sentence about the subjects/animals and methods.
  3. in the abstract please give p-values and number of cases
  4. In the abstract please re-modify: We believe.. is not scientific language
  5. In the whole Please check the writing of Latin names. Enteropathogenic and Enterohemorrhagic are not Latin names. Do you mean Akkermansia spp. and do you mean certain species in Bifidobacteria spp.? Or are you meaning the genus in general, all the species in the genus, unknown species of the genus or one particular/certain species of the genus? please clarify the meaning and sentences. If you are meaning all possible member in the genes, many cases English-Latin names like salmonella, pseudomonas, campylobacteria, bifidobacterial etc. with a small capital letter. Please correct!
  6. You were using coli O157:H7. please add the definition enterohemorrhagic in the title, abstract and introduction
  7. lines 93-5. Please identify the bacterial genome DNA extracting kit and how the host own cells were depleted and please clarify ‘ All DNA ‘
  8. lines 96 & 110. The original sequences and Primitive sequences : do you mean primers and probes as well as PCR protocols?
  9. lines 96-7. Explain what you have really done in detail: “Take the negative control group as 1, and calculate relative contents of other groups. “
  10. line 99. NF-κB pathway is more than IL-6, TNF-α and IL-1β -please correct and do not use abbreviation not explained before
  11. line 98 and 104. There is ‘inflammatory levels’ in both titles please correct or merge the 2.2 and 2.3
  12. lines 105-6. please add the reference ‘Health condition of colon can be reflected indirectly by measuring mucin genes expression level, for example, MUC2 and Occludin were often used to detect mucin content in colon.’ or remove it.
  13. The whole Please correct the terminology. Since you measure mucin expression please remove from the words (abstract, intro, m&m and results) like intestinal barrier function and replaced with the terminology which you have done. in the discussion you may claim that mucin is related to gut barrier function and explain how.
  14. please explain in discussion or other convenient place how you can use 16S rDNA sequence data to predict bacterial their function (line 128)
  15. Line 128-9. please explain and give refences how you are able to apply TAX4Fun algorithm to 16S rDNA sequence data
  16. Lines 112-33. whole genome shotgun sequencing (WGS), produces metagenomic data not 16S rDNA amplicon sequencing. please be precise and correct
  17. lines 143-4. add the reference: ‘Caco-2 cell is a kind of clonal human colonic adenocarcinoma cell, whose structure and function are similar to differentiated intestinal epithelial cell.’
  18. lines 166-170. Re-place in here all data analyses including statistical analyses and data analyses from 16S rDNA data under 2.7 title
  19. lines 173-5. Please add the used references ‘As a principle type of EHEC, coli O157 has its unique virulence gene, which do not exist in most microorganism in our environment, hence, number of characteristic virulence gene Stx1 was used to evaluate the body's infection situation’
  20. line 175 re-define /re-write ’the body's infection situation’
  21. How the colonization rate was defined. Please add.
  22. lines 182-3. References? Re-place into the discussion ‘2'-FL has similar results with many other prebiotics, but the specific cause needs further investigation on its microflora and intestinal environment’. Moreover, define probiotics and do not use old fashion terminology like microflora . see also the same words in the whole manuscript e.g. line 219,252, 258 etc
  23. The whole results section. Please add the calculated p-values. Do not say ‘highly significant difference’ if you do not present significant p-values.
  24. line 197. please do not give statements without the references in the results. Add the ref or omit
  25. lines 202-217. terminology: Factors -> use mediators / cytokines
  26. The whole results section in the text and tables / figures. add the used statistical methods. As well as add the explanations of abbreviations into tables and figures
  27. m&m. where the histological methods are explained?
  28. lines 205-8. terminology: please define ‘Inflammation expression’ and ‘stomach intervention’  and use precise words
  29. lines 205-10. ‘This illustrated .. ‘ Re-write or remove into the discussion, please.
  30. line 228-232. please correct see comment 5
  31. lines 234-47. Please give references and explanation why species levels information is given from 16S data or where this data is come from? and add the p-values
  32. line 240-7. What is ‘Cancers ascended’ -? Add pathway -words of terminology and p-values
  33. Figure 4. Species: is that correct?
  34. line 313. re-locate into the methods and define which manufacturer: ‘Cell vitality was determined under manufacturer's instructions’
  35. The limitations of methods and experiments are not discussed as well as the fact that microbiota of mice differs highly from the microbiota of humans. Please add the limitation paragraph

Minor comments

  1. the whole text and tables/figures: open the abbreviations like MUC2, IL-x, TNF, DEPC, UPGMA, SCFA, VPA, DB-FFAP
  2. line 136. so? could you replaced it with more scientific word
  3. line 149. 6 x 105 cells: correct

Round 2

Reviewer 1 Report

The manuscript now is much improved.  However, authors still need to address the following points -

The authors should data on dose-response studies.  Also, they should highlight that the strain used lacked the virulent gene.  Host tailors its response based on the challenge it faces- pathogen type, virulence, dose etc.  Thus, results should be qualified with clear statement that response to a highly virulent toxin-producing E coli might be different.  Regarding the mechanisms, there is already plenty of information on the plausible mechanisms that should be acknowledged.

Reviewer 2 Report

Dear Sirs

The manucript has proven, thank you. Although it would have helped me a lot if the the line numbers were correct and corresponding the answers as well as there would be clean corrected version of the manucript too and not only the tracked version. 

MAJOR mistag/ uncorrected issues:

1. Please re- correct the latin names at least in the discussion there are many Bifidobacteria and Akkermansia using italics without the spp. please consult microbiologist to read through your manucript

2. In the response of 9. Please shortly explain the relative content of each sample lines x 

3. In the response 16: please do not use old fashion sound flora word but use preferentially intestinal microbiome
